# Distribution of human papillomavirus genotypes by severity of cervical lesions in HPV screened positive women from the ESTAMPA study in Latin America

Rita Mariel Correa[1]*, Armando Baena[2], Joan Valls[2,3], María Celeste Colucci[1], Laura Mendoza[4], Maryluz Rol[2], Carolina Wiesner[5], Annabelle Ferrera[6], María Dolores Fellner[1], Joaquín Víctor González[1], Jorge Alejandro Basiletti[1], Pamela Mongelos[4], Mercedes Rodriguez de la Peña[7], Agustina Saino[8], Elena Kasamatsu[4], Carlos Velarde[9], Ninoska Macavilca[9], Sandra Martinez[5], Gino Venegas[10], Alejandro Calderón[11], Guillermo Rodriguez[12], Hernán Barrios[4], Rolando Herrero[13], Maribel Almonte[2], María Alejandra Picconi[1], on behalf of the ESTAMPA Study Group[¶]

1 Servicio Virus Oncogénicos, Laboratorio Nacional y Regional de Referencia de HPV, Instituto Nacional de Enfermedades Infecciosas- ANLIS "Dr. Malbrán", Buenos Aires, Argentina, 2 Early Detection, Prevention & Infections Branch, International Agency for Research on Cancer, World Health Organization, Lyon, France, 3 Centro de Investigación Biomédica en Red de Cáncer (CIBERONC), Madrid, Spain, 4 Instituto de Investigaciones en Ciencias de la Salud, Universidad Nacional de Asunción, San Lorenzo, Paraguay, San Lorenzo, Paraguay, 5 Instituto Nacional de Cancerología, Bogotá, Colombia, 6 Instituto de Investigaciones en Microbiología, Universidad Nacional Autónoma de Honduras, Tagucigalpa, Honduras, 7 Servicio Ginecología, Hospital Nacional "Prof. Posadas", El Palomar, Argentina, 8 Sección Histología, Servicio Anatomía Patológica, Hospital Nacional "Prof. Posadas", El Palomar, Argentina, 9 Instituto Nacional de Enfermedades Neoplásicas, Surquillo, Perú, 10 Clínica Anglo Americana, Lima, Perú, 11 Caja Costarricense de Seguro Social, Región Pacífico Central, San José, Costa Rica, 12 Comisión Honoraria de Lucha contra el Cáncer, Montevideo, Uruguay, 13 Agencia Costarricense de Investigaciones Biomédicas, San José, Costa Rica

¶ Membership of the ESTAMPA study group is provided in the Acknowledgments.
* correeamariel01@gmail.com

**Data Availability Statement:** All relevant data are within the paper and its Supporting Information files.

## Abstract

The proportion of HPV16 and 18-associated cervical cancer (CC) appears rather constant worldwide (≥70%), but the relative importance of the other HR-HPV differs slightly by geographical region. Here, we studied the HPV genotype distribution of HPV positive Latin American (LA) women by histological grade, in a sub-cohort from the ESTAMPA study; we also explored the association of age-specific HPV genotypes in severe lesions. Cervical samples from 1,252 participants (854 ≤CIN1, 121 CIN2, 194 CIN3 and 83 CC) were genotyped by two PCRs-Reverse Blotting Hybridization strategies: i) Broad-Spectrum General Primers 5+/6+ and ii) PGMY9/11 PCRs. HPV16 was the most frequently found genotype in all histological grades, and increased with the severity of lesions from 14.5% in ≤ CIN1, 19.8% in CIN2, 51.5% in CIN3 to 65.1% in CC (p < 0.001). For the remaining HR-HPVs their frequency in CC did not increase when compared to less severe categories. The nonavalent vaccine HR-types ranked at the top in CC, the dominant ones being HPV16 and HPV45. HR-HPV single infection occurs, respectively, in 57.1% and 57.0% of ≤CIN1 and CIN2, increasing to 72.2% and 91.6% in CIN3 and CC (p<0.001). No association between age and HPV type was observed in CC, although the risk of HPV16 infection in CIN3 cases

**Funding:** This research has been partially supported by grants from IARC (grant number CRA-PRI), the Instituto Nacional del Cáncer (Argentina) and Agencia Nacional de Promoción de la Investigación, el Desarrollo Tecnológico y la Innovación (Argentina) (grant number 0364-2016), all awarded to MAP. The funders had no role in study design, data collection and analysis, decision to publish, or preparation of the manuscript.

**Competing interests:** The authors have declared that no competing interests exist.

increased with age. Results confirm the relevance of HPV16 in the whole clinical spectrum, with a strong rise of its proportion in CIN3 and cancer. This information will be relevant in evaluating the impact of HPV vaccination, as a baseline against which to compare genotype changes in HPV type-specific distribution as vaccinated women participate in screening in LA region. Likewise, these data may help select the best HPV testing system for HPV-based efficient, affordable, and sustainable screening programmes.

## Introduction

Human papillomaviruses (HPVs) are a large and diverse group of viruses comprising more than 230 fully characterized types, even though new HPV types are still being reported [1, 2]. Among them, about 40 genotypes are known to infect the anogenital and aerodigestive tracts, distinguishing between high risk (HR)-HPV and low risk (LR)-HPV types in relation to their oncogenic potential [3, 4].

The International Agency for Research on Cancer (IARC) classifies twelve HPV genotypes as carcinogenic (HR-HPV) (Group 1: HPVs 16, 18, 31, 33, 35, 39, 45, 51, 52, 56, 58, 59), while HPV68 is probably carcinogenic (probable HR-HPV) (Group 2A) and HPVs 26, 53, 66, 67, 69, 70, 73 and 82 are possibly carcinogenic (possible HR-HPV) (Group 2B); the so-called LR-HPV types (HPVs 6, 11, 40, 42, 43, 44, 55, 61, 81, 83) strongly evidence that they do not cause cancer (Group 4) [4–6]. It has been firmly established that persistent HR-HPV infection is the main cause of cervical cancer (CC), HPV16 and HPV18 are associated to more than 70% of the cases [7–11].

Even though a marked reduction has been observed worldwide over the last decades, CC incidence and mortality is still high in low and middle-income countries (LMICs). Particularly, in Latin American and Caribbean countries (LAC), 59,500 new CC cases and 31,500 CC deaths are estimated every year [12] and are still considered a serious health problem.

In the last 15 years, HPV vaccination programmes have been progressively introduced in high-income countries but also in most LAC countries, especially using the quadrivalent vaccine (4v)(Gardasil) [13]; however, efforts to establish well organized screening programmes for CC prevention are still needed since even for vaccinated women some form of screening will be required [14, 15].

HPV DNA testing is currently replacing cytology in CC screening and recent WHO recommendations suggest its implementation as primary test in both, screen and treat and screen, triage and treat approaches [15–18]. Currently, several laboratory assays have been clinically validated to detect HR-HPV, reporting high sensitivity and reproducibility and allowing to extend of the screening interval [19]. Therefore, an increasing number of countries have switched from cytology to HPV-based screening, or have decided to introduce this change in the near future [15, 20, 21].

The ESTAMPA study (Spanish acronym for "EStudio multicéntrico de TAMizaje y triaje del cáncer cervico uterino con pruebas de virus PApiloma humano") is a multicentre cross-sectional study aimed at evaluating the performance of different triage techniques to detect cervical precancerous lesions, and at informing how to implement HPV-based screening programmes in Latin American (LA) women [22].

Whereas the proportion of HPVs 16 and 18-associated CC appears rather constant worldwide, the relative importance of the other carcinogenic HPV types differs slightly by geographical region [7, 8, 10]. The HPV type distribution in high-grade cervical lesions (HSIL), i.e. cervical intraepithelial neoplasia grades 2 and 3 (CIN2 and CIN3), is not entirely representative of those in CC. HPVs 16, 18 and 45 are more common in CC than in high-grade cervical

lesions, although the other HR-HPV types are more common in CIN2 and 3 compared with CC [8, 23]. Moreover, different HPV genotypes have different natural histories, and it has become increasingly important to identify which genotypes are most indicative of high risk to develop precancerous lesions [24, 25].

Results on the genotype distribution of HPV are essential for estimating the impact of vaccines on CC and to adequately plan, implement and improve HPV-based cervical screening programmes. Here, we present data on HPV genotype distribution in a sub-cohort of women testing HPV positive, selected from ESTAMPA according to their histopathological diagnoses; we also explored the association of age-specific HPV genotypes with severe cervical lesions.

## Materials and methods

### Study population

Details of the ESTAMPA study design, methods and protocol are published elsewhere [22]. For the current analysis, 27,558 participants from six study centres (Argentina, Colombia, Paraguay, Honduras, Costa Rica, and Uruguay), were recruited from May 2013 to October 2018, and were considered (Fig 1).

The participants signed an informed consent that included details on the background, study procedures, risks and benefits, statement of confidentiality, specimen use and study staff to contact.

At the time of recruitment (first study visit), three consecutive samples of cervical cells were obtained from all participants: dry swab (Dacron) for oncoprotein testing, $1^{st}$ cytobrush (PreservCyt Vial 1) for HPV testing and potential morphological triage techniques (i.e. dual-stained cytology) and $2^{nd}$ cytobrush (PreservCyt Vial 2) that was aliquoted and frozen for further molecular triage testing.

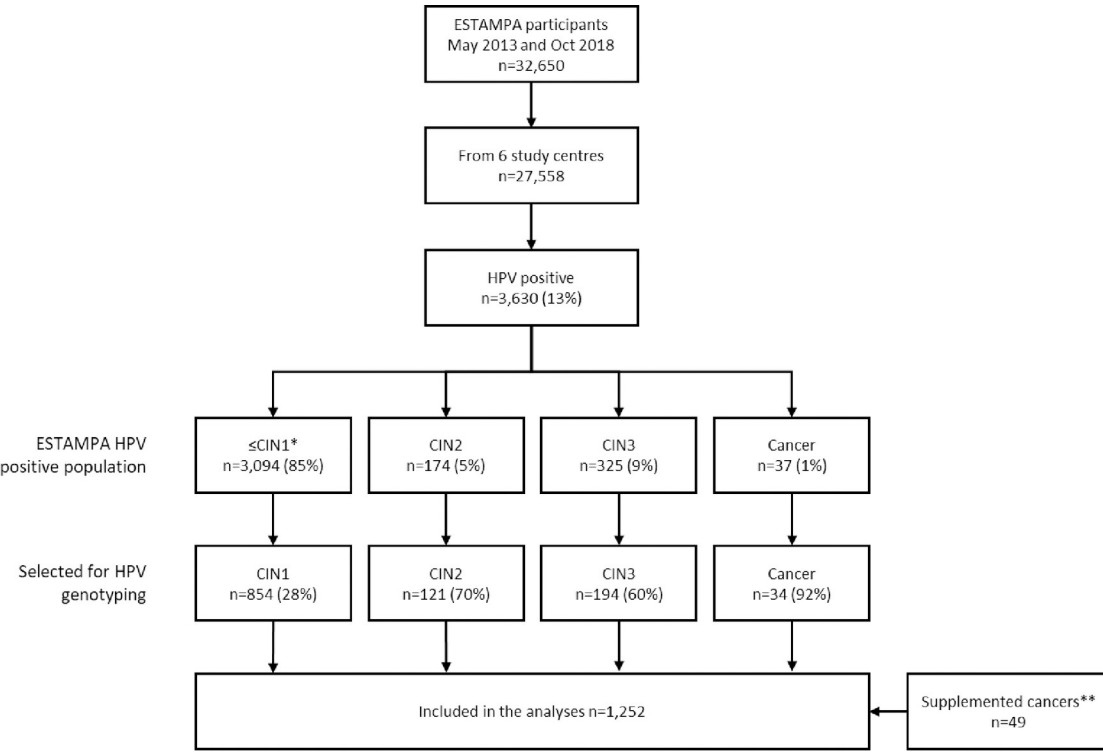

**Fig 1. Study population.** *Includes colposcopy negative, biopsy negative or CIN1**Cancers supplemented from oncologic clinics of 2 countries hosting ESTAMPA.

Samples and aliquots were stored at each centre until they were used or were transferred to a central hub for specific testing or to IARC for long-term storage.

Women with a positive HPV test were referred for colposcopy with biopsies, disease ascertainment and treatment as needed. For this ancillary study, four independent series of HPV-positive ESTAMPA participants were selected by convenience based on their known diagnosis, including 854 ≤CIN1 (negative colposcopy/biopsy or CIN1), 121 CIN2, 194 CIN3 and 34 CC. In addition, 49 CC cases recruited at Instituto Nacional de Enfermedades Neoplásicas and Clínica San Gabriel (Perú), and Instituto Nacional de Cancerología (Colombia) were also included to enrich the CC series. Biological samples from these participants were processed for HPV genotyping at the Regional HPV Reference Laboratory (Global HPV LabNet) from Argentina.

## DNA extraction from cervical samples

Total DNA was extracted from 1000-μl aliquots of exfoliated cervical cells (Vial 2), stored at -20˚C. These samples had been collected in ThinPrep PreservCyt Solution (Hologic, Bedford, MA, USA) using a cervical Rovers® Cervex-Brush® (Broom-like collection devices) at the time of the recruitment in each centre; this brush allows for simultaneous harvesting of ecto-cervical, endocervical and transformation zone cells with a single device.

The extraction was performed using commercial columns (Qiagen) on a robotic system (QIAcube system, Qiagen), following manufacturer's instructions; for every ten samples, a tube containing only medium was included as a control for contamination. DNAs were stored at −20˚C until use.

## HPV detection and genotyping

HPV genotyping was performed by two PCR techniques combined with Reverse Blotting Hybridization (RBH) using type-specific biotynilated probe sets: i) Broad-Spectrum General Primers 5+/6+ PCR (BSGP) and ii) PGMY9/11 (CHUV), which allow to amplify highly conserved 140 bp and 450 bp fragments of the HPV-L1 gene, respectively [26–28]. Results from both techniques were combined in order to detect as many HPV genotypes as possible, since the performance of each PCR used to separately identify genotypes is different.

Briefly, two aliquots (5 μL) of the purified DNA were PCR amplified (total reaction volume 50 μL) with AmpliTaq polymerase (Thermo Fisher Scientific, Waltham, MA, USA) and biotinylated primers. Positive (0.1 pg/mL of HPV16 full length genomic DNA in a plasmid) and negative controls (dH2O) were used to assess the sensitivity of PCR and detect contaminating HPV DNA in reagents.

RBH was performed as described [29, 30]; denatured PCR products were allowed to hybridize with specific oligonucleotide probes corresponding to: i) BSGP-PCR primers, 36 HPV genotypes (HPVs: 6, 11, 16, 18, 26, 31, 33, 34, 35, 39, 40, 42, 43, 44, 45, 51, 52, 53, 54, 55, 56, 57, 58, 59, 61, 66, 68, 70, 71, 72, 73, 81, 82, 83, 84 and HPV89), and ii) CHUV primers, 32 HPV genotypes (HPVs: 6, 11, 16, 18, 26, 31, 33, 34, 35, 39, 40, 42, 43, 44, 45, 51, 52, 53, 54, 55, 56, 57, 58, 59, 66, 68, 69, 70, 73, 82, 83 and 84). Amplicons were immobilized on a Biodyne C membrane (Pall corporation, Port Washington, NY, USA) using a Miniblotter MN45 (Immunetics, Cambridge, MA, USA).

The hybrids formed were treated with alkaline phosphatase-streptavidin conjugate (GE HealthCare) and substrate (ECL Detection Reagents, GE HealthCare), resulting in a chemiluminescent product subsequently detected by exposure to autoradiography film.

The Regional HPV Reference Laboratory from Argentina annually participates in the external control of HPV genotyping (HPV DNA typing proficiency panel), provided by the

International Papillomavirus Reference Centre, which contributes reliability to the results obtained in this study [31].

### Ethical considerations

The ESTAMPA protocol was approved by the Ethics Committee of the International Agency for Research on Cancer (IEC Project 12–27-A7), the Pan American Health Organization (PAHO) Ethical Committee and Ethical Committees in each of the study participating centres. The study is considered minimal risk as the procedures are standard practice in cervical cancer screening programmes [22]. Material Transfer Agreements were signed as needed.

### Statistical analysis

Samples were considered positive for genotyping, if they were positive for one or more genotypes, by either technique. In case of multiple infections, each genotype was counted independently; however, when grouping genotypes, a hierarchical rule was applied.

The frequency and prevalence of HPV genotypes were computed within each histological group. In addition, five HPV genotype risk-based groups were defined as follows: (1) HPV16/18 (carcinogenic types), (2) other HR-HPV (including other high-risk carcinogenic HPVs 31, 33, 35, 39, 45, 51, 52, 56, 58, 59 types, probably carcinogenic HPV68 type and possibly carcinogenic HPV66 type; all of them in the HPV screening techniques cocktails); (3) possibly HR-HPV (including other possibly carcinogenic HPVs 26, 34, 53, 69, 70, 73, 82 types), (4) LR-HPV (including low-risk HPVs 6, 11, 40, 42, 43, 44, 54, 55, 57, 61, 71, 72, 81, 83, 84, 89 types) and (5) negative (no HPV genotypes detected). The five groups were mutually exclusive; participants were included in a single risk-based group following a hierarchical rule described as follows: (i) HPV16/18 included the participants positive for HPV16 and/or HPV18 (both in single and multiple infections), (ii) other HR-HPV included those positive for any genotype in the group without HPV16/18, (iii) possibly HR-HPV included those positive for any genotype in the group without HPV16/18 or other HR-HPV infections, (iv) LR-HPV included those participants positive for any genotype in the group without any of the previous infections. Furthermore, the frequency and prevalence of participants who were positive for three or more, only two, only one or none of the 14 genotypes in the HPV screening techniques cocktails (i.e. HPVs 16, 18, 31, 33, 35, 39, 45, 51, 52, 56, 58, 59, 66, 68) were also computed within the histological diagnoses. Moreover, prevalence was also calculated for all pairs of co-occurring HR-HPV genotypes (91 pairs from 14 different genotypes). In addition, the frequency and prevalence of HR-HPV genotypes within each histologic group for single infections were also computed, thus restricting the analysis to those women whose sample was positive for only one HPV genotype (666 out of 1,252 participants). Exact 95% confidence intervals were computed for prevalence interval estimation using a binomial distribution. Finally, to assess the association of individual HPV genotypes and HPV risk-based groups positivity with age (considering age groups 30–39, 40–49 and 50–65) within histological group, risk-ratios were computed (considering the 30–39 age group as reference) and computing 95% confidence intervals using a normal approximation (Wald). Trend tests for proportions were also performed when appropriate using a chi-squared test. All analysis were performed using the R statistical software and a threshold for significance at $\alpha = 0.05$.

## Results

The study population description according to histological diagnoses and age is detailed in Table 1, showing a larger proportion of women aged 40 and above in cancers when compared to CIN2 or CIN3.

**Table 1. Distribution of age in by histological group in HPV screened positive women.**

| Age, years | ≤CIN1* | CIN2 | CIN3 | Cancer |
|---|---|---|---|---|
|  | n = 854 | n = 121 | n = 194 | n = 83 |
| Mean (SD) | 43.8 (9.51) | 39.6 (8.59) | 39.9 (8.01) | 45.8 (8.69) |
| 30–39 | 337 (39.5%) | 75 (62%) | 106 (54.6%) | 22 (26.5%) |
| 40–49 | 251 (29.4%) | 29 (24%) | 56 (28.9%) | 28 (33.7%) |
| 50–64 | 266 (31.1%) | 17 (14%) | 32 (16.5%) | 33 (39.8%) |

Mean (SD: standard deviation) and absolute frequency (percentage within groups) shown.

*≤CIN1 includes colposcopy negative, biopsy negative or CIN1.

Table 2 describes the prevalence of HPV genotype risk-based groups within histological diagnoses among HPV screened positive women. A significant increasing trend of the HPV16/18 group prevalence with histological grade was observed (ranging from 20% in ≤ CIN1, to 30.6% in CIN2, 56.7 in CIN3 and 71.1% in CC, p-value for trend <0.001); conversely, the prevalence of other HR-HPV groups decreased with histological grade (ranging from 58%, to 51.2%, 38.7 and 26.5%, respectively, p-value for trend <0.001). The prevalence of other possibly HR-HPV and LR-HPV genotypes was much lower, and mostly found in women with ≤ CIN1 and CIN2 diagnoses (4.1% and 7.4%, respectively for possibly HR-HPV and 3.2% and 5.8%, respectively for LR-HPV). The majority of women with no HPV genotype detected were among those with a ≤ CIN1 diagnosis. No substantial differences were observed in the prevalence of HPV genotype risk-based groups when distinguishing between colposcopy/biopsy negative and CIN1 diagnoses (S1 Table).

Despite a higher genotypic diversity in women with ≤ CIN1 and CIN2 diagnoses compared to CIN3 and CC cases (Fig 2), HPV16 was the most prevalent genotype in all histological groups, prominently in women with cancer, observing a significant upward trend with histological grade (ranging from 14.5%, 19.8%, 51.5% to 65.1% in ≤ CIN1, CIN2, CIN3 and cancer, respectively, p-value for trend p<0.001) (Table 3). Furthermore, among women with CC, the prevalence of the remaining HPV genotypes was below 9%; among women with CIN3, HPV31 was the second most prevalent genotype, followed by HPV52, HPV58 and HPV33 (12.9%,

**Table 2. Prevalence of HPV genotype risk-based groups within histological diagnoses in HPV screened positive women.**

|  | ≤CIN1 | CIN2 | CIN3 | Cancer | p-value |
|---|---|---|---|---|---|
|  | n (% 95%IC) | n (% 95%IC) | n (% 95%IC) | n (% 95%IC) |  |
| HPV16/18 | 171 (20% 17.4–22.9) | 37 (30.6% 22.5–39.6) | 110 (56.7% 49.4–63.8) | 59 (71.1% 60.1–80.5) | p<0.001 |
| Other HR-HPV | 495 (58% 54.6–61.3) | 62 (51.2% 42–60.4) | 75 (38.7% 31.8–45.9) | 22 (26.5% 17.4–37.3) | p<0.001 |
| P Possibly HR-HPV | 35 (4.1% 2.9–5.7) | 9 (7.4% 3.5–13.7) | 3 (1.5% 0.3–4.5) | 1 (1.2% 0–6.5) | p = 0.09 |
| LR-HPV | 27 (3.2% 2.1–4.6) | 7 (5.8% 2.4–11.6) | 0 (0% 0–1.9) | 1 (1.2% 0–6.5) | p = 0.049 |
| Negative | 126 (14.8% 12.4–17.3) | 6 (5% 1.8–10.5) | 6 (3.1% 1.1–6.6) | 0 (0% 0–4.3) | p<0.001 |
|  | 854 (100%) | 121 (100%) | 194 (100%) | 83 (100%) |  |

Number of positive participants, prevalence (%) and 95% confidence intervals for each HPV risk-based group shown within each histological diagnosis. Genotypes included in five risk-based groups as follows: (1) HPV16/18 (carcinogenic types); (2) other HR-HPV (including other carcinogenic HPVs 31, 33, 35, 39, 45, 51, 52, 56, 58, 59 types), probably carcinogenic HPV68 type and possibly carcinogenic HPV66 type, all twelve of them in the HPV screening techniques cocktails (3) possibly HR-HPV (including other possibly carcinogenic HPVs 26, 34, 53, 69, 70, 73, 82 types); (4) LR-HPV (including LR-HPVs 6, 11, 40, 42, 43, 44, 54, 55, 57, 61, 71, 72, 81, 83, 84, 89 types); and (5) negative (none of the genotypes detected). Participants assigned to one single risk-based group considering a hierarchy from HPV16/18, other HR-HPV, possibly HR-HPV to LR-HPV in relation to the positivity of the corresponding genotypes. P-value from a trend test for proportions to assess the association of the prevalence with the histological grade for each HPV risk-based groups shown.

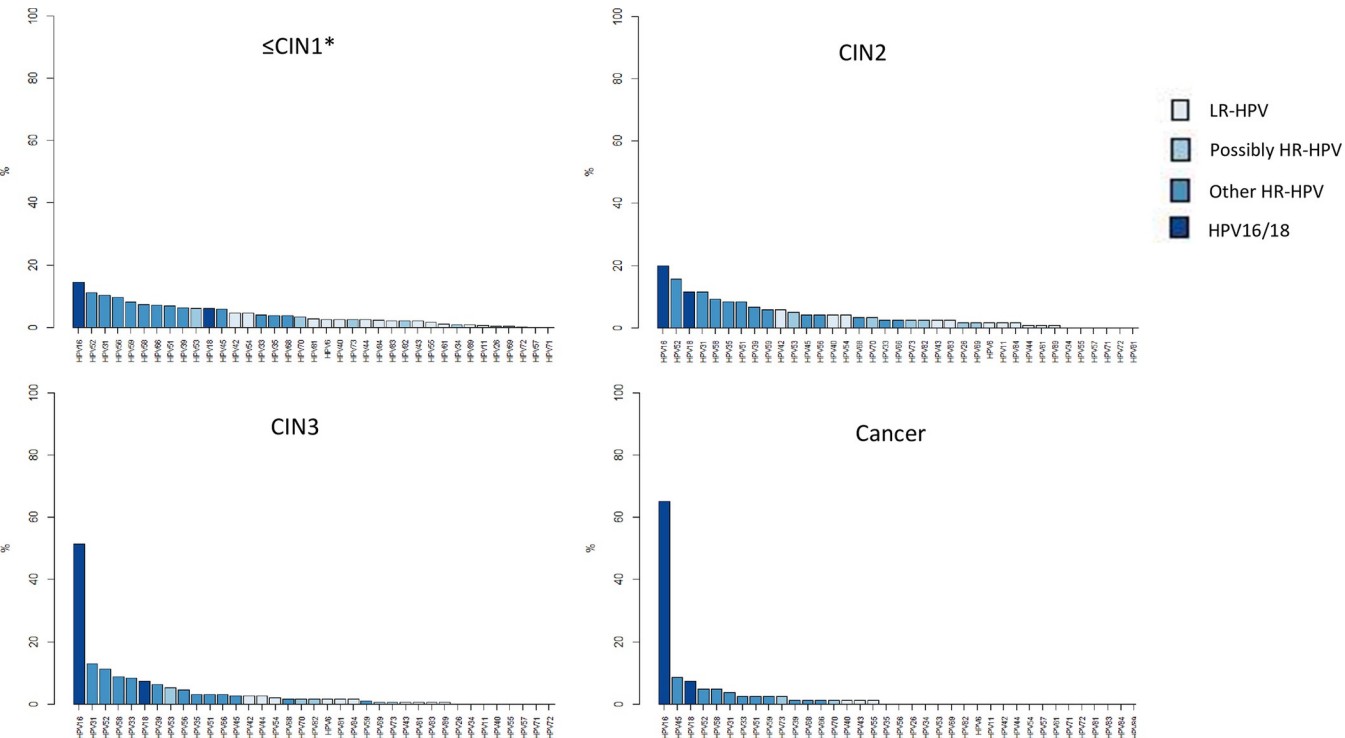

**Fig 2. Distribution of HPV genotype prevalence within histological diagnoses in HPV screened positive women.** Prevalence (%) shown as bars within each histological group. Colours represent different HPV genotypes risk-based groups: (1) HPV16/18 (carcinogenic types), (2) other HR-HPV (including other carcinogenic HPV31/33/35/39/45/51/52/56/58/59 types, probably carcinogenic HPV68 type and possibly carcinogenic HPV66 type; all twelve of them in the HPV screening techniques cocktails); (3) possibly HR-HPV (including other possibly carcinogenic HPV26/ 34/53/69/70/73/82 types), (4) LR-HPV (including low-risk HPV6/11/40/42/43/44/54/55/57/61/71/72/81/83/84/89 types) and (5) negative (none of the genotypes detected).

11.3%, 8.8% and 8.2%, respectively); among women with CIN2, HPV52 was the second most prevalent genotype, followed by HPV18, HPV31, HPV58, HPV35 and HPV51 (15.7%, 11.6%, 11.6%, 9,1%, 8.3% and 8.3%, respectively); and among women with a ≤ CIN1 diagnosis, HPV52 was also the second most prevalent genotype, followed by HPV31, HPV56, HPV59, HPV58, HPV 66 and HPV51 (11.1%, 10.3%, 9.5%, 8.2%, 7.3%, 7% and 6.9%, respectively). Most of the other possibly carcinogenic and LR-HPV genotypes had a low prevalence and were almost not detected in cancer cases (S2 Table). Similar results were found when considering those participants with single infections (S3 Table).

Table 4 describes the prevalence of a single HR-HPV infection and co-infections involving two, three or more HR-HPV genotypes within histological diagnoses and presents the ten most frequent double HR-HPV infections. The majority of women had single infections, notably in CC cases (91.6%) and CIN3 diagnoses (72.2%). Infections with two HR-HPV genotypes were detected in less than 20% and those with three or more genotypes in around 5% of ≤ CIN1, CIN2 or CIN3 cases. Overall, HPV16 with HPVs 31, 52, 39, 18, 66, 58, 59 and HPV52 with HPV18/31/56 were the most frequent HR-HPV co-infections (complete list provided in S4 Table).

Table 5 explores the association of HPV16 and HPV genotype risk-based groups with age in histological diagnoses. The risk of HPV16 infection in CIN3 cases increased with age (p-value for trend = 0.03); women aged 50 to 65 showed higher prevalence than those aged 30 to 39 (RR 1.45 95% CI 1.05–2.01). In addition, a downward trend in LR-HPV genotypes was observed in women with ≤ CIN1diagnosis (p-value for trend = 0.02). No other association

**Table 3. Prevalence of high-risk (HR) HPV genotype infections within histological diagnoses in HPV screened positive women.**

|  | ≤CIN1 n = 854 | CIN2 n = 121 | CIN3 n = 194 | Cancer n = 83 |
|---|---|---|---|---|
|  | n (% 95%IC) | n (% 95%IC) | n (% 95%IC) | n (% 95%IC) |
| **HR-HPV genotypes (Group 1)** |  |  |  |  |
| HPV16 | 124 (14.5% 12.2–17.1) | 24 (19.8% 13.1–28.1) | 100 (51.5% 44.3–58.8) | 54 (65.1% 53.8–75.2) |
| HPV18 | 51 (6% 4.5–7.8) | 14 (11.6% 6.5–18.7) | 14 (7.2% 4–11.8) | 6 (7.2% 2.7–15.1) |
| HPV31 | 88 (10.3% 8.3–12.5) | 14 (11.6% 6.5–18.7) | 25 (12.9% 8.5–18.4) | 3 (3.6% 0.8–10.2) |
| HPV33 | 34 (4% 2.8–5.5) | 3 (2.5% 0.5–7.1) | 16 (8.2% 4.8–13) | 2 (2.4% 0.3–8.4) |
| HPV35 | 32 (3.7% 2.6–5.2) | 10 (8.3% 4–14.7) | 6 (3.1% 1.1–6.6) | 0 (0% 0–4.3) |
| HPV39 | 54 (6.3% 4.8–8.2) | 8 (6.6% 2.9–12.6) | 12 (6.2% 3.2–10.6) | 1 (1.2% 0–6.5) |
| HPV45 | 50 (5.9% 4.4–7.6) | 5 (4.1% 1.4–9.4) | 5 (2.6% 0.8–5.9) | 7 (8.4% 3.5–16.6) |
| HPV51 | 59 (6.9% 5.3–8.8) | 10 (8.3% 4–14.7) | 6 (3.1% 1.1–6.6) | 2 (2.4% 0.3–8.4) |
| HPV52 | 95 (11.1% 9.1–13.4) | 19 (15.7% 9.7–23.4) | 22 (11.3% 7.2–16.7) | 4 (4.8% 1.3–11.9) |
| HPV56 | 81 (9.5% 7.6–11.7) | 5 (4.1% 1.4–9.4) | 9 (4.6% 2.1–8.6) | 0 (0% 0–4.3) |
| HPV58 | 62 (7.3% 5.6–9.2) | 11 (9.1% 4.6–15.7) | 17 (8.8% 5.2–13.7) | 4 (4.8% 1.3–11.9) |
| HPV59 | 70 (8.2% 6.4–10.2) | 7 (5.8% 2.4–11.6) | 2 (1% 0.1–3.7) | 2 (2.4% 0.3–8.4) |
| **Probably HR-HPV genotype (Group 2A)** |  |  |  |  |
| HPV68 | 31 (3.6% 2.5–5.1) | 4 (3.3% 0.9–8.2) | 3 (1.5% 0.3–4.5) | 1 (1.2% 0–6.5) |
| **Possible HR-HPV genotype (Group 2B)** |  |  |  |  |
| HPV66 | 60 (7% 5.4–9) | 3 (2.5% 0.5–7.1) | 6 (3.1% 1.1–6.6) | 1 (1.2% 0–6.5) |

Number of positive participants, prevalence (%) and 95% confidence intervals shown within each histological group. Results shown for individual HPV genotypes (irrespective of positivity for other genotypes).

**Table 4. Number of high-risk (HR)-HPV genotype infections within histological diagnoses in HPV screened positive women and most common HR-HPV co-infections.**

|  | ≤CIN1 | CIN2 | CIN3 | Cancer |
|---|---|---|---|---|
|  | n (% 95%IC) | n (% 95%IC) | n (% 95%IC) | n (% 95%IC) |
| **Number of HR-HPV genotypes** |  |  |  |  |
| None | 188 (22% 19.3–24.9) | 22 (18.2% 11.8–26.2) | 9 (4.6% 2.1–8.6) | 2 (2.4% 0.3–8.4) |
| One | 488 (57.1% 53.7–60.5) | 69 (57% 47.7–66) | 140 (72.2% 65.3–78.3) | 76 (91.6% 83.4–96.5) |
| Two | 138 (16.2% 13.8–18.8) | 24 (19.8% 13.1–28.1) | 36 (18.6% 13.3–24.8) | 4 (4.8% 1.3–11.9) |
| Three or more | 40 (4.7% 3.4–6.3) | 6 (5% 1.8–10.5) | 9 (4.6% 2.1–8.6) | 1 (1.2% 0–6.5) |
| Total number of women | 854 (100%) | 121 (100%) | 194 (100%) | 83 (100%) |
| **HR-HPV genotypes pairs** |  |  |  |  |
| HPV16 and HPV31 | 10 (1.2% 0.6–2.1) | 3 (2.5% 0.5–7.1) | 4 (2.1% 0.6–5.2) | 0 (0% 0–4.3) |
| HPV16 and HPV52 | 5 (0.6% 0.2–1.4) | 1 (0.8% 0–4.5) | 7 (3.6% 1.5–7.3) | 0 (0% 0–4.3) |
| HPV52 and HPV18 | 7 (0.8% 0.3–1.7) | 2 (1.7% 0.2–5.8) | 3 (1.5% 0.3–4.5) | 0 (0% 0–4.3) |
| HPV52 and HPV31 | 8 (0.9% 0.4–1.8) | 3 (2.5% 0.5–7.1) | 1 (0.5% 0–2.8) | 0 (0% 0–4.3) |
| HPV16 and HPV39 | 6 (0.7% 0.3–1.5) | 1 (0.8% 0–4.5) | 4 (2.1% 0.6–5.2) | 0 (0% 0–4.3) |
| HPV16 and HPV18 | 4 (0.5% 0.1–1.2) | 1 (0.8% 0–4.5) | 4 (2.1% 0.6–5.2) | 1 (1.2% 0–6.5) |
| HPV16 and HPV66 | 7 (0.8% 0.3–1.7) | 0 (0% 0–3) | 2 (1% 0.1–3.7) | 1 (1.2% 0–6.5) |
| HPV52 and HPV56 | 8 (0.9% 0.4–1.8) | 0 (0% 0–3) | 2 (1% 0.1–3.7) | 0 (0% 0–4.3) |
| HPV16 and HPV58 | 5 (0.6% 0.2–1.4) | 1 (0.8% 0–4.5) | 2 (1% 0.1–3.7) | 1 (1.2% 0–6.5) |
| HPV16 and HPV59 | 5 (0.6% 0.2–1.4) | 2 (1.7% 0.2–5.8) | 2 (1% 0.1–3.7) | 0 (0% 0–4.3) |

Number of positive participants, prevalence (%) and 95% confidence intervals shown by the number of genotypes detected (none, one, two or three or more; mutually exclusive groups). Computations for the ten most frequent pairs of genotypes detected also shown.

**Table 5. Association between HPV16 and HPV risk-based groups infections with age within histological diagnoses in HPV screened positive women.**

| | n | HPV16 n(%) | HPV16 RR (95%CI) | HPV16/18 n(%) | HPV16/18 RR (95%CI) | Other HR-HPV n(%) | Other HR-HPV RR (95%CI) | Possibly HR-HPV n(%) | Possibly HR-HPV RR (95%CI) | LR-HPV n(%) | LR-HPV RR (95%CI) |
|---|---|---|---|---|---|---|---|---|---|---|---|
| ≤CIN1 | | | | | | | | | | | |
| All | 854 | 124 (14.5%) | | 171 (20%) | | 495 (58%) | | 35 (4.1%) | | 27 (3.2%) | |
| 30–39 | 337 | 45 (13.4%) | 1 | 62 (18.4%) | 1 | 202 (59.9%) | 1 | 12 (3.6%) | 1 | 16 (4.7%) | 1 |
| 40–49 | 251 | 33 (13.1%) | 0.98 (0.65–1.5) | 46 (18.3%) | 1 (0.71–1.41) | 146 (58.2%) | 0.97 (0.85–1.11) | 13 (5.2%) | 1.45 (0.68–3.13) | 7 (2.8%) | 0.59 (0.25–1.41) |
| 50–65 | 266 | 46 (17.3%) | 1.3 (0.89–1.89) | 63 (23.7%) | 1.29 (0.94–1.76) | 147 (55.3%) | 0.92 (0.8–1.06) | 10 (3.8%) | 1.06 (0.46–2.41) | 4 (1.5%) | 0.32 (0.11–0.94)* |
| CIN2 | | | | | | | | | | | |
| All | 121 | 24 (19.8%) | | 37 (30.6%) | | 62 (51.2%) | | 9 (7.4%) | | 7 (5.8%) | |
| 30–39 | 75 | 16 (21.3%) | 1 | 23 (30.7%) | 1 | 38 (50.7%) | 1 | 6 (8%) | 1 | 6 (8%) | 1 |
| 40–49 | 29 | 6 (20.7%) | 0.97 (0.42–2.23) | 10 (34.5%) | 1.12 (0.61–2.06) | 15 (51.7%) | 1.02 (0.67–1.55) | 2 (6.9%) | 0.86 (0.18–4.03) | 1 (3.4%) | 0.43 (0.05–3.43) |
| 50–65 | 17 | 2 (11.8%) | 0.55 (0.14–2.18) | 4 (23.5%) | 0.77 (0.31–1.93) | 9 (52.9%) | 1.04 (0.63–1.72) | 1 (5.9%) | 0.74 (0.09–5.71) | 0 (0%) | - |
| CIN3 | | | | | | | | | | | |
| All | 194 | 100 (51.5%) | | 110 (56.7%) | | 75 (38.7%) | | 3 (1.5%) | | 0 (0%) | |
| 30–39 | 106 | 48 (45.3%) | 1 | 56 (52.8%) | 1 | 47 (44.3%) | 1 | 0 (0%) | - | 0 (0%) | - |
| 40–49 | 56 | 31 (55.4%) | 1.22 (0.89–1.67) | 33 (58.9%) | 1.12 (0.84–1.48) | 19 (33.9%) | 0.77 (0.5–1.17) | 2 (3.6%) | - | 0 (0%) | - |
| 50–65 | 32 | 21 (65.6%) | 1.45 (1.05–2.01)* | 21 (65.6%) | 1.24 (0.91–1.69) | 9 (28.1%) | 0.63 (0.35–1.15) | 1 (3.1%) | - | 0 (0%) | - |
| Cancer | | | | | | | | | | | |
| All | 83 | 54 (65.1%) | | 59 (71.1%) | | 22 (26.5%) | | 1 (1.2%) | | 1 (1.2%) | |
| 30–39 | 22 | 15 (68.2%) | 1 | 17 (77.3%) | 1 | 4 (18.2%) | 1 | 0 (0%) | - | 1 (4.5%) | - |
| 40–49 | 28 | 20 (71.4%) | 1.05 (0.72–1.52) | 22 (78.6%) | 1.02 (0.75–1.37) | 6 (21.4%) | 1.18 (0.38–3.67) | 0 (0%) | - | 0 (0%) | - |
| 50–65 | 33 | 19 (57.6%) | 0.84 (0.56–1.27) | 20 (60.6%) | 0.78 (0.55–1.12) | 12 (36.4%) | 2 (0.74–5.41) | 1 (3%) | - | 0 (0%) | - |

Number of positive participants and prevalence (%) for each genotype shown within each histological diagnosis. Risk ratio (RR) with 95% confidence interval shown considering as reference group women aged 30–39. Genotypes included in each group as follows: (1) HPV16/18 (carcinogenic types), (2) other HR-HPV (including other carcinogenic HPVs 31, 33, 35, 39, 45, 51, 52, 56, 58, 59 types, probably carcinogenic HPV68 type and possibly carcinogenic HPV66 type; all twelve of them in the HPV screening techniques cocktails; (3) possibly HR-HPV (including other possibly carcinogenic HPVs 26, 34, 53, 69, 70, 73, 82 types) and (4) LR-HPV (including LR-HPVs 6, 11, 40, 42, 43, 44, 54, 55, 57, 61, 71, 72, 81, 83, 84, 89 types). Participants assigned to one single risk-based group considering a hierarchy from HPV16/18, other HR-HPV, possibly HR-HPV to LR-HPV in relation to the positivity of the corresponding genotypes. Risk ratios and confidence intervals not computed in case of null frequencies.

* Statistically significant confidence interval.

between HPV risk-based genotypes with age was observed. S5 Table presents further assessment for selected HR-HPV individual genotypes (HPVs 18, 31, 45, 52); no significant trend was noted.

## Discussion

This study provides detailed information about the HPV genotypes' frequency distribution among Latin American HPV positive women from a sub-cohort of the ESTAMPA study, enriched with CC samples.

HPV16 was by far the most frequently detected type in all grades, with a minor variation between ≤ CIN1 (14.5%) and CIN2 (19.8%), but remarkably rising in CIN3 (51.5%) and CC (65.1%). These data are in line with previous reports from the global meta-analysis of HPV-types' distribution, which also reports HPV16 as the most common type, whose presence steadily increases with the severity of histological changes, from normal cytology (20.4%), CIN1 (27.6%) CIN2 (39.8%) and CIN3 (58.2%), to its peak in CC (62.6%) [8]. The persistence of HPV16 infection has been established as the main indicator and predictor of CIN3 development [32, 33]. The fact that our data showed no difference in the HPV16 frequency between CIN3 and CC, although differences were noted with less severe lesions, and this effect for HPV16 was not seen in other HR-HPV analysed, further evidences that HPV16 positive women would need a stricter clinical management.

In our series, women diagnosed with ≤CIN1 and CIN2, showed the greatest genotypes diversity, both HR and LR-HPV (35 genotypes identified), whose individual frequencies did not exceed 25%, as has been already described [34, 35]; this confirms the widespread effect of these viruses in the sexually active population.

On the other hand, in CIN3 and CC, the viral spectrum was much more limited, detecting almost exclusively HR-HPVs, as was previously described [7–11]. The inverse correlation between HPV diversity and progressive disease is also consistent with the findings on 1,518 cervical biopsies ranging from negative to CIN3 in the ATHENA trial (Addressing The Need for Advanced HPV diagnostics) [33].

There seems to be no clear consensus on the ranking of the next oncogenic genotypes in the clinical spectrum, beyond HPV16. In the present study, other carcinogenic types were detected more frequently in intermediate cervical diagnoses than in CC. In CIN3, HPV31 ranked second after HPV16, followed by HPV 52, 58, 33, 18 and 39 with frequencies under 13%. Our results are in line with those obtained by Arbyn et al. in a large collection of samples from Belgium, indicating in precancerous lesions, the presence of HPV16, followed by HPV31 and HPV52 [36]. A worldwide meta-analysis by Guan et al. identified HPV31 (11.7%) as the second most common type in CIN3, close to the rate (12.9%) [8] found in this study. We found that HPV45 ranked tenth, almost the same as the report by Kjaer SK et al. in Danish women (eleventh place) [37], although in disagreement with the number found by Guan et al., where it advanced to the seventh place [8].

In this LA series, our results for CC agree with previous research, the most common HPV types being the seven HR-HPVs included in the nonavalent vaccine, which ranked in descending order as follows: HPV16, 45, 18, 52, 58, 31 and 33 [7, 10, 38]. These findings add evidence to the major benefit of the 9v HPV vaccine against CC and might help to extend the prevention to other HPV-related cancers, in a larger number of people [39, 40]. Furthermore, it is well known that HPV16, 18 and 45 are associated with endocervical cancers, whose precursor lesions are not so easily seen at colposcopy; if these infections remain persistent, a more careful exploration of the endocervical canal may be needed, especially in older women (>30 years).

Some countries have started CC screening using HPV detection, however the role of identifying the individual HR-types is unresolved and debated internationally [25]. Currently, some approved HPV tests with concurrent partial HPV genotyping separately identify HPVs 16 and 18, a few of them also individually identify HPVs 45 and 31 [19], and new systems are extending this capacity to other HR-HPVs. Our results are in line with recent studies that maintain that the risk of progression differs substantially by HPV type, and conclude that HPV typing, yielding information at least for HPV16 (probably HPV18, and possibly HPV45), is a useful and worthy aspect of a state-of-the-art HPV test used for primary cervical cancer screening and management [25]. Consensus should be reached on which types are essential to better identify women at real high risk of CIN3 and cancer [24, 41].

According to our results, the implementation of a high coverage prophylactic vaccine could reduce the number of CIN3 and CC (caused by HPVs 16 and 18) by up to 56.7% and 71.1%, respectively. Since 2v/4v HPV vaccines, the only ones currently available in the immunization programmes of the LAC region, show cross reactivity with non-vaccine HPV-types, particularly HPVs 31 and 45; an extra prevention against precancer and cancer can also be considered [40]. These findings are also supported by the results of the early impact of HPV vaccination study in sexually active adolescent girls from Argentina, showing a significant drop not only in HPVs 16 and 18 prevalence, but also in HPVs 31 and 45 [30]. Other reports have optimistically predicted the benefit of HPV-vaccination on the overall number of cervical abnormalities [39, 40] and more recently it was reported that the HPV immunization programme has successfully almost eliminated CC in vaccinated women from Sweeden and UK [42, 43]. However, as expected, our data indicate that vaccination would have a lower impact on ≤CIN2 since HPVs 16 and 18, in these clinical categories, are found in 30.6% in CIN2 and 20% in ≤ CIN1 respectively. Moreover, a fraction of HPVs 16 and 18 positive samples contained other HPV types as well and it is not clear yet to what extent the vaccination would have an impact on these cases [44].

Our series showed a decreasing frequency of other HR-HPV types (HPVs 31/ 33/ 35/ 39/ 45/ 51/ 52/ 56/ 59/ 66/ 68) from ≤CIN1 (58%) to CC (26.5%), agreeing with previous data from the LA region [10, 38]. Although these types substantially contribute to the burden of infection, they are not as prevalent in CIN3 and CC, and are responsible for less than 7% of CC worldwide [45].

Multiple infections are a topic of interest, because the implications of their presence remains unclear. They have been frequently observed in all the spectrum of cervical lesions [9, 46, 47]; Schmitt and colleagues found that the occurrence of multiple HPV infections did not affect the risk of a lesion being high- or low-grade [48], and Wentzensen and colleagues reported no association between disease status and the number of genotypes detected in a woman [49]. Previous studies showing increased risk of CIN with multiple HPV infections had few CIN2+ cases, and were restricted to younger women [50], a subgroup known to harbour a larger number of HPV infections [51]. In our study, a greater percentage of HR-HPV single infections occurs in CC cases compared to ≤ CIN1, CIN2 and CIN3. The marked minority of multiple HR-HPV infections identified in CC agree with former publications based on histological microdissection, indicating that any individual cervical lesion is produced by a single virus and reducing the pathogenetic importance of HR-HPV interactions [52]. Therefore, the fact that fewer multiple infections were detected in the more severe cervical lesions is consistent with the ecological principles of competitive exclusion and carcinogenesis hallmarked by clonal expansion and evolution of transformed cells [53, 54].

In the double infection group, our data showed that the occurrence of HPV16 coinfection with other HR-HPV was less common in CIN1 than in CIN3 lesions, similar to that reported by previous research [55]. Some authors have suggested that phylogenetically related HPV genotypes have the tendency to cluster together (i.e. alpha 9) [47] while others have proposed that coinfecting HPV genotypes occur at random and lead to cervical disease independently [50]. In our work, the most prevalent double infection involved genotypes belonging to alpha 9 + alpha9, alpha9 +alpha 7, and alpha9 +alpha 6 species, which would seem to be closer to this last statement; however, the limited number of multiple infections of our study did not allow us to further deepen this topic.

Some authors have reported an age-specific pattern of HR-HPV distribution with larger proportions of precancerous lesions and CC in older women attributed to HR-HPV types different than HPV16/18, but results are still controversial [56–60]. More recent data from Finland showed that HPV type distribution in HSIL+ was distinctly polarised by age with

HPV16/18 attributed disease being markedly more prevalent in women aged <30 [58]. The only published information on this issue in Latin American women comes from a population-based study conducted in Guanacaste, Costa Rica, which investigated the HPV types detected in CIN 2/3 and CC by age (from 18 to ≥56 years old). It reported that 80% of CIN3+ were associated with HPV16 among women 18–26 years old compared to only 32% among women older than 55 (p trend = 0.018) [61]. Although our data did not show a significant association between age and any type or groups of types in severe lesions, except for HPV16, an increasing trend could be observed with a peak in the range 50–65 years old range compared to those aged 30–39 for CIN3, suggesting this lesion in older women where mainly caused by HPV16. In addition, the exclusion of women younger than 30 years old in our study, among other limitations in terms of statistical power given the limited sample size, may explain the lack of statistical significance of our observations. Finally, it is also worth mentioning that our study suggests that a second peak in HPV16 infection is seen in women aged above 50+years, which is consistent with the findings of Porras et al.; this allows us to speculate on a particular pattern for LAC women, unlike what is found in other populations.

We found 11% of samples (mostly in women diagnosed with ≤CIN1) that were HPV-positive at screening (predominantly by HC2 test), but not in HPV genotyping. It should be considered that the PCR-RBH genotyping was performed in the vial 2, which could have had fewer cells than the vial 1 in which the HPV screening test was carried out. However, we cannot be ruled out that these discrepant results could be false positives or represent cross-reactions with non-carcinogenic HPV types. This HC2 test drawback has also been reported by others [44, 62]. Moreover, two studies have found that by increasing the cut-off level from the FDA-approved of 1.0 to 2 or 3 relative light units (RLU/Co ratio) greatly decreases the HPV prevalence among women with normal cytology or mild cytological abnormalities and reduces screening false positive results [44, 63].

Among the main strengths of this study is the increased knowledge on the HPV genotypes' distribution in a large series of samples with histological diagnosis, in the context of a well-organized screening and triage trial in Latin American women. However, the study also has limitations. First, it was a cross sectional study performed in a sub-cohort selected by convenience to obtain similar number of samples in each of the diagnostic categories and cannot be claimed to represent the wider screening at the population level; and second, having chosen to run the study in women eligible for screening by HPV testing (aged ≥30), it was not possible to recover data from younger women.

In conclusion, the results confirm the relevance of HPV16 in the whole clinical spectrum of cervical lesions, with a strong rise of its proportion in CIN3+; particularly, in CC, the top ranking HPV types are the seven HR-HPVs included in the 9v HPV vaccine, with HPV16, HPV18, HPV45 prevaling. This information will be relevant in evaluating the impact of HPV vaccination, as a baseline against which to compare genotype changes in HPV type-specific distribution in associated lesions as vaccinated women participate in screening. Evaluations of the overall reduction in infections by vaccine types, and changes in other related alpha types would be some of the main points to evaluate and would help to maintain and improve immunization programmes. Likewise, these data may help to evaluate screening strategies, aiming to choose the best HPV testing system and establish HPV-based efficient, affordable, and sustainable screening programmes.

The huge heterogeneity of the populations and resources available at global level prioritises the importance of regional studies to evaluate specific determining factors and make the best public policy decisions, in harmony with the WHO strategy to achieve the elimination of CC [64].

## Supporting information

**S1 Table. Prevalence of HPV genotype risk-based groups within histological diagnoses in HPV screened positive women.** Number of positive participants, prevalence (%) and 95% confidence intervals for each HPV risk-based group shown within each histological diagnosis. Genotypes included in five risk-based groups as follows: (1) HPV16/18 (carcinogenic types), (2) other HR-HPV (including other carcinogenic HPV31/33/35/39/45/51/52/56/58/59 types, probably carcinogenic HPV68 type and possibly carcinogenic HPV66 type; all twelve of them in the HPV screening techniques cocktails); (3) possibly HR-HPV (including other possibly carcinogenic HPV26/ 34/53/69/70/73/82 types), (4) LR-HPV (including low-risk HPV6/11/ 40/42/43/44/54/55/57/61/71/72/81/83/84/89 types) and (5) negative (none of the genotypes detected). Participants assigned to one single risk-based group considering a hierarchy from HPV16/18, other HR-HPV, possibly HR-HPV to LR-HPV in relation to the positivity of the corresponding genotypes. P-value from a trend test for proportions to assess the association of the prevalence with the histological grade for each HPV risk-based groups shown.
(DOCX)

**S2 Table. Prevalence of possibly carcinogenic and low risk (LR)-HPV genotype infections within histological diagnoses in HPV screened positive women.** Number of positive participants, prevalence (%) and 95% confidence intervals shown within each histological group. Results shown for individual HPV genotypes (irrespective of positivity for other genotypes).
(DOCX)

**S3 Table. Prevalence of high risk (HR)-HPV genotype single infections within histological diagnoses in HPV screened positive women.** Number of positive participants, prevalence (%) and 95% confidence intervals shown within each histological group. Results shown for individual HPV genotypes (irrespective of positivity for other genotypes).
(DOCX)

**S4 Table. Prevalence of high risk (HR)-HPV genotype double infections within histological diagnoses in HPV screened positive women.** Number of participants, prevalence (%) and 95% confidence intervals shown within each histological group for all possible pairs (91 pairs) of HR genotypes ordered by their overall detection rate.
(DOCX)

**S5 Table. Prevalence of HPV18, HPV31, HPV45 and HPV52 genotypes within histological diagnoses overall and by age in HPV screened positive women.** Number of positive participants and prevalence (%) for each genotype shown within each histological diagnosis. Risk ratio (RR) with 95% confidence interval shown considering as reference group women aged 30–39. Risk ratios and confidence intervals not computed in case of null frequencies. P-values from a trend test for proportions shown.
(DOCX)

## Acknowledgments

The team is thankful to all women who participated in this study.

ESTAMPA Study Group: Valeria Mariel Padín, Servicio Virus Oncogénicos, Laboratorio Nacional y Regional de Referencia de HPV, Instituto Nacional de Enfermedades Infecciosas-ANLIS "Dr. Malbrán", Buenos Aires, Argentina; Juan Mural, Alicia Inés Figueredo, María Alejandra Borelli, Paula Gabriela Falabella, Natalia Gabriela Kaznowiecki, Servicio Ginecología, Hospital Nacional "Prof. Posadas", El Palomar, Argentina; Sandra García, Adrián Moreno, Julieta Guyot, Melisa Alejandra Lencioni, Nerina Cattarin, Mariano Quinteiro, Servicio

Anatomía Patológica, Hospital Nacional "Prof. Posadas", El Palomar, Argentina; Ana Peraza González, Hospital México, Caja Costarricense de Seguro Social, San José, Costa Rica; Marisol Brizuela Hernandez, Centro Nacional de Citologías, Caja Costarricense de Seguro Social, San José, Costa Rica; Emmanuel González Solano, Hospital Dr. Enrique Baltodano, Caja Costarricense de Seguro Social, San José, Costa Rica; Carlos Sosa Quesada, Hospital Monseñor Víctor Manuel Sanabria, Caja Costarricense de Seguro Social, San José, Costa Rica; Rolando Pinastel Monet and Ricci Fernández Hidalgo, Area de Salud Barranca, Caja Costarricense de Seguro Social, San José, Costa Rica; Johanna González Arias, Área de Salud San Rafael de Puntarenas, Caja Costarricense de Seguro Social, San José, Costa Rica; Yessi Cabrera, Instituto de Investigaciones en Microbiología, Universidad Nacional Autónoma de Honduras, Tagucigalpa, Honduras; María Isabel Rodriguez, Malvina Paez, Amalia Castro, Claudia Centurión, Romina Báez, Jaime Vester, Instituto de Investigaciones en Ciencia de la Salud, Universidad Nacional de Asunción, Paraguay; Ana Soilán, Hospital Nacional de Itauguá y Hospital Materno Infantil de San Lorenzo, Ministerio de Salud Pública y Bienestar Social, Asunción, Paraguay; Marina Ortega, Hospital Nacional de Itaugua, e Instituto Nacional del Cáncer, Ministerio de Salud Pública y Bienestar Social, Asunción, Paraguay; Carmen Cristaldo, Hospital Nacional de Itaugua, Ministerio de Salud Pública y Bienestar Social, Asunción, Paraguay.

Lead author for this group; Valeria Mariel Padín. Email: oncovir@anlis.gob.ar

Where authors are identified as personnel of the International Agency for Research on Cancer / World Health Organization, the authors alone are responsible for the views expressed in this article and they do not necessarily represent the decisions, policy or views of the International Agency for Research on Cancer / World Health Organization.

## Author Contributions

**Conceptualization:** Rita Mariel Correa, Rolando Herrero, Maribel Almonte, María Alejandra Picconi.

**Data curation:** Armando Baena, Joan Valls.

**Formal analysis:** Armando Baena, Joan Valls.

**Funding acquisition:** Rolando Herrero, Maribel Almonte, María Alejandra Picconi.

**Investigation:** Rita Mariel Correa, María Celeste Colucci, Laura Mendoza, Maryluz Rol, María Dolores Fellner, Joaquín Víctor González, Jorge Alejandro Basiletti, Mercedes Rodriguez de la Peña, Agustina Saino, Elena Kasamatsu, María Alejandra Picconi.

**Methodology:** Rita Mariel Correa, María Celeste Colucci, Joaquín Víctor González, Jorge Alejandro Basiletti.

**Project administration:** Rolando Herrero, Maribel Almonte, María Alejandra Picconi.

**Resources:** Rita Mariel Correa, Laura Mendoza, Carolina Wiesner, Annabelle Ferrera, Pamela Mongelos, Mercedes Rodriguez de la Peña, Agustina Saino, Elena Kasamatsu, Carlos Velarde, Ninoska Macavilca, Sandra Martinez, Gino Venegas, Alejandro Calderón, Guillermo Rodriguez, Hernán Barrios.

**Software:** Armando Baena, Joan Valls.

**Supervision:** María Alejandra Picconi.

**Validation:** Maryluz Rol.

**Visualization:** María Alejandra Picconi.

**Writing – original draft:** Rita Mariel Correa.

**Writing – review & editing:** Rita Mariel Correa, Armando Baena, Joan Valls, María Celeste Colucci, Laura Mendoza, Maryluz Rol, María Dolores Fellner, Rolando Herrero, Maribel Almonte, María Alejandra Picconi.

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
