## [Decision Letter · Decision Letter 0]

10 May 2022

PONE-D-22-10439Distribution of human papillomavirus genotypes by severity of cervical lesions in HPV screened positive women from the ESTAMPA study in Latin America.PLOS ONE

Dear Dr. Correa,

Thank you for submitting your manuscript to PLOS ONE. After careful consideration, we feel that it has merit but does not fully meet PLOS ONE’s publication criteria as it currently stands. Therefore, we invite you to submit a revised version of the manuscript that addresses the points raised during the review process.Please submit your revised manuscript by Jun 24 2022 11:59PM. If you will need more time than this to complete your revisions, please reply to this message or contact the journal office at plosone@plos.org. Please include the following items when submitting your revised manuscript:A rebuttal letter that responds to each point raised by the academic editor and reviewer(s). You should upload this letter as a separate file labeled 'Response to Reviewers'.A marked-up copy of your manuscript that highlights changes made to the original version. You should upload this as a separate file labeled 'Revised Manuscript with Track Changes'.An unmarked version of your revised paper without tracked changes. You should upload this as a separate file labeled 'Manuscript'.If applicable, we recommend that you deposit your laboratory protocols in protocols.io to enhance the reproducibility of your results. Protocols.io assigns your protocol its own identifier (DOI) so that it can be cited independently in the future. For instructions see: https://journals.plos.org/plosone/s/submission-guidelines#loc-laboratory-protocols. Additionally, PLOS ONE offers an option for publishing peer-reviewed Lab Protocol articles, which describe protocols hosted on protocols.io. Read more information on sharing protocols at https://plos.org/protocols?utm_medium=editorial-email&utm_source=authorletters&utm_campaign=protocols.

We look forward to receiving your revised manuscript.

Kind regards,

Maria Lina Tornesello

Academic Editor

PLOS ONE

Journal Requirements:

2.Please review your reference list to ensure that it is complete and correct. If you have cited papers that have been retracted, please include the rationale for doing so in the manuscript text, or remove these references and replace them with relevant current references. Any changes to the reference list should be mentioned in the rebuttal letter that accompanies your revised manuscript. If you need to cite a retracted article, indicate the article’s retracted status in the References list and also include a citation and full reference for the retraction notice.

"The authors are obliged to all gynaecologists, pathologists, nurses, and technician who have contributed to the collection of samples analysed in this study. This research has been partially supported by IARC (CRA-PRI), Instituto Nacional del Cáncer (Argentina) and Agencia Nacional de Promoción de la Investigación, el Desarrollo Tecnológico y la Innovación (Argentina) (Grant No. 0364-2016 to MAP)."

"Yes - Rol: María Alejandra Picconi (MAP) PI

- Colaborative research agreement (IARC CRA-PRI) to MAP 

-Instituto Nacional del Cancer (Argentina) 2016 to MAP

- Agencia Nacional de Promocion de la Investigacion , el Desarrollo Tecnologico y la Innovacion (Grant No. 0364-2016 to MAP)

5. One of the noted authors is a group or consortium ESTAMPA Study Group. In addition to naming the author group, please list the individual authors and affiliations within this group in the acknowledgments section of your manuscript. Please also indicate clearly a lead author for this group along with a contact email address.

Reviewers' comments:

Reviewer's Responses to Questions

**Comments to the Author**

1. Is the manuscript technically sound, and do the data support the conclusions?

Reviewer #1: Yes

Reviewer #2: Yes

2. Has the statistical analysis been performed appropriately and rigorously? 

Reviewer #1: Yes

Reviewer #2: Yes

3. Have the authors made all data underlying the findings in their manuscript fully available?

Reviewer #1: Yes

Reviewer #2: Yes

4. Is the manuscript presented in an intelligible fashion and written in standard English?

Reviewer #1: Yes

Reviewer #2: Yes

5. Review Comments to the Author

Reviewer #1: The study reported HPV genotyping results from 1252 LA women with positive HPV test and cervical biopsy findings. These data may help to evaluate screening strategies and the impact of HPV vaccination.

Some comments:

1. The participants include 854 cases of ≤CIN1. I agree it is easy to analyze the data to reduce the group numbers. However, it is interesting to know the HPV genotyping results in cases with negative biopsy and CIN1, separately. I suggest authors divide the 854 cases into two groups for analysis.

2. Table 2 includes 5 groups, HPV16/18, other HR-HPV, Possibly HR-HPV, LR-HPV, negative. Many cases showed multiple HPV infections. The last row shows the numbers: 854, 121, 194, 83, all with 100%. I assume HPV16/18 group includes the cases with some other HPV infection (multiple); other HR-HPV group with no HPV 16/18; LR-HPV with no any other HR-HPV et al. Suggest the authors to explain details.

3. Based the study design all the precipitants were HPV positive. I do not understand why these 138 women (126, 6, 6) had negative HPV testing in table 2.

4. Overall, more than 20% cases had multiple HPV infections. I assume that Table 3 showed the prevalence of any hrHPV Genotype (single or multiple). It is great if the authors list also the prevalence of single hrHPV genotype, which can demonstrate individual HPV effect on cervical lesions.

Reviewer #2: In the study ”Distribution of human papillomavirus genotypes by severity of cervical lesions in HPV screened positive women from the ESTAMPA study in Latin America.”, the authors are presenting the results of HPV genotyping of an impressive number of women.

The used genotyping HPV assays are not commercially, clinically validated, but participating at HPV DNA typing proficiency panel, and IARC collaboration also, are a proof of high-quality assessment of samples.

The results are necessary to sustain in this area also, the replacement of Pap smear with HPV genotyping for cervical cancer screening, which together with vaccination could lead to CC elimination.

6. PLOS authors have the option to publish the peer review history of their article (what does this mean?). If published, this will include your full peer review and any attached files.

Reviewer #1: No

Reviewer #2: **Yes: **Ramona Gabriela Ursu

---

## [Author Response · Author response to Decision Letter 0]

7 Jul 2022

REBUTTAL LETTER

Dear Editor and Reviewer 1,

 First, the authors would like to thank the Editor and Reviewer1 for the time spent evaluating our manuscript, and the valuable comments made that helped improve it.

 Responses to Reviewer 1:

1- The participants include 854 cases of ≤CIN1. I agree it is easy to analyze the data to reduce the group numbers. However, it is interesting to know the HPV genotyping results in cases with negative biopsy and CIN1, separately. I suggest authors divide the 854 

The meta-analysis reported by Guan et al showed no clear difference in the HR-type distribution between HPV-positive normal cytology and CIN1, also that the CIN1 diagnoses had limited usefulness for risk-stratification of HPV-positive women; so CIN1 would predominantly be a “proxy” for HPV infection rather than a true cancer precursor (Guan et al; Int J Cancer, 2012). On the other hand, the genotypes that we identified in samples of women with “negative colposcopy/biopsy” and “CIN1” were similar. For such reasons, we initially put together these two categories in the same group (≤CIN1), and we would prefer to keep it that way.

Nevertheless, we found the reviewer's suggestion interesting, so we decided to include a supplementary table in the support information (Table S1) where Table 2 appears with the ≤CIN1 group broken down into 2 groups: “negative colposcopy/ biopsy” and “CIN1”.

 Therefore, in RESULTS, a new sentence was added:

Lines 210-212: “No substantial differences were observed in the prevalence of HPV genotype risk-based groups when distinguishing between colposcopy/biopsy negative and CIN1 diagnoses (Table S1)”

2 - Table 2 includes 5 groups, HPV16/18, other HR-HPV, Possibly HR-HPV, LR-HPV, negative. Many cases showed multiple HPV infections. The last row shows the numbers: 854, 121, 194, 83, all with 100%. I assume HPV16/18 group includes the cases with some other HPV infection (multiple); other HR-HPV group with no HPV 16/18; LR-HPV with no any other HR-HPV et al. Suggest the authors to explain details.

The reviewer interpreted correctly. To further clarify the description already included in MATERIAL AND METHODS (Statistical analysis), we added a phrase (highlighted in yellow in the revised manuscript) as follows:

Lines 175- 181: “The five groups were mutually exclusive; participants were included in a single risk-based group following a hierarchical rule described as follows: (i) HPV16/18 included the participants positive for HPV16 and/or HPV18 (both in single and multiple infections) (ii) other HR-HPV included those positive for any genotype in the group without HPV16/18, (iii) possibly HR-HPV included those positive for any genotype in the group without HPV16/18 or other HR-HPV infections (iv) LR-HPV included those participants positive for any genotype in the group without any of the previous infections.” 

3- Based on the study design all the precipitants were HPV positive. I do not understand why these 138 women (126, 6, 6) had negative HPV testing in table 2.

As the Reviewer says, this study included women who had been HPV positive by a screening test (HC2 or Cobas HPV) performed on the first PreservCyt Vial 1 (first cytobrush obtained). In a second cytobrush, obtained subsequent to the previous one (PreservCyt Vial 2), genotyping was performed by PCR-RBH, which allows the identification of 37 HPV genotypes (alone or coinfecting). In 138 samples, no HPV genotype was detected. This has been previously described by other authors (Leinonen MK et al, Br J Cancer. 2013; Hesselink AT et al., J Clin Microbiol. 2006).

The DISCUSSION presents the possible reasons that justify this result, as follows:

Lines 388-396: “We found 11% of samples (mostly in women diagnosed with ≤CIN1) that were HPV-positive at screening (predominantly by HC2 test), but not in HPV genotyping. It should be considered that the PCR-RBH genotyping was performed in vial 2, which could have had fewer cells than vial 1 in which the HPV screening test was performed. However, it cannot be ruled out that these discrepant results could be false positives or represent cross-reactions with non-carcinogenic HPV types. Others also reported this drawback of the HC2 test [44, 62]. Moreover, two studies have found that increasing the cut-off level from the FDA-approved of 1.0 to 2 or 3 relative light units (RLU/Co ratio) greatly decreases the HPV prevalence among women with normal cytology or mild cytological abnormalities and reduces screening’s false positive results [44, 63].

4- Overall, more than 20% cases had multiple HPV infections. I assume that Table 3 showed the prevalence of any HR-HPV Genotype (single or multiple). It is great if the authors list also the prevalence of single HR-HPV genotype, which can demonstrate individual HPV effect on cervical lesions. 

We have accepted the Reviewer´s suggestion. We added a new phrase in MATERIAL AND METHODS (Statistical analysis) and RESULTS as follows: 

 Line 185-187: “In addition, the frequency and prevalence of HR-HPV genotypes within each histologic group for single infections were also computed, thus restricting the analysis to those women whose sample was positive for only one HPV genotype (666 out of 1,252 participants)”

Lines 238-239: “Similar results were found when considering the participants with a single infection”.

We also added a supplementary table (Table S3) in the support information section, which shows individual HPV effect on the analyzed groups.

Responses to Editor:

1. Please ensure that your manuscript meets PLOS ONE's style requirements.

 The style requirements were met.

2. Please review your reference list to ensure that it is complete and correct

References were revised.

3. Please update your submission to use the PLOS LaTeX template

According to instructions sent to us by Glenn Jackson (Production Editor), our revised manuscript was created using Word and submitted in PDF format (Revised Manuscript with Track Changes).

"The authors are obliged to all gynaecologists, pathologists, nurses, and technician who have contributed to the collection of samples analysed in this study. This research has been partially supported by IARC (CRA-PRI), Instituto Nacional del Cáncer (Argentina) and Agencia Nacional de Promoción de la Investigación, el Desarrollo Tecnológico y la Innovación (Argentina) (Grant No. 0364-2016 to MAP)." 

It was removed from Acknowledgments Section 

 We note that you have provided funding information that is not currently declared in your Funding Statement. However, funding information should not appear in the Acknowledgments section or other areas of your manuscript

Currently, your Funding Statement reads as follows: 

"Yes - Rol: María Alejandra Picconi (MAP) PI

- Colaborative research agreement (IARC CRA-PRI) to MAP 

-Instituto Nacional del Cancer (Argentina) 2016 to MAP

- Agencia Nacional de Promocion de la Investigacion , el Desarrollo Tecnologico y la Innovacion (Grant No. 0364-2016 to MAP)

Current Funding Statement is OK

5. One of the noted authors is a group or consortium ESTAMPA Study Group. In addition to naming the author group, please list the individual authors and affiliations within this group in the acknowledgments section of your manuscript. Please also indicate clearly a lead author for this group along with a contact email address.

ESTAMPA Study Group members were listed in the ACKNOWLEDGMENTS section as follows: 

LINES 419-440: “The team is thankful to all women who participated in this study.

ESTAMPA Study Group: Valeria Mariel Padín, Servicio Virus Oncogénicos, Laboratorio Nacional y Regional de Referencia de HPV, Instituto Nacional de Enfermedades Infecciosas- ANLIS "Dr. Malbrán", Buenos Aires, Argentina; Juan Mural, Alicia Inés Figueredo, María Alejandra Borelli, Paula Gabriela Falabella, Natalia Gabriela Kaznowiecki, Servicio Ginecología, Hospital Nacional “Prof. Posadas”, El Palomar, Argentina; Sandra García, Adrián Moreno, Julieta Guyot, Melisa Alejandra Lencioni, Nerina Cattarin, Mariano Quinteiro, Servicio Anatomía Patológica, Hospital Nacional “Prof. Posadas”, El Palomar, Argentina; Ana Peraza González, Hospital México, Caja Costarricense de Seguro Social, San José, Costa Rica; Marisol Brizuela Hernandez, Centro Nacional de Citologías, Caja Costarricense de Seguro Social, San José, Costa Rica; Emmanuel González Solano, Hospital Dr. Enrique Baltodano, Caja Costarricense de Seguro Social, San José, Costa Rica; Carlos Sosa Quesada, Hospital Monseñor Víctor Manuel Sanabria, Caja Costarricense de Seguro Social, San José, Costa Rica; Rolando Pinastel Monet and Ricci Fernández Hidalgo, Área de Salud Barranca, Caja Costarricense de Seguro Social, San José, Costa Rica; Johanna González Arias, Área de Salud San Rafael de Puntarenas, Caja Costarricense de Seguro Social, San José, Costa Rica; Yessi Cabrera, Instituto de Investigaciones en Microbiología, Universidad Nacional Autónoma de Honduras, Tagucigalpa, Honduras; María Isabel Rodriguez, Malvina Paez, Amalia Castro, Claudia Centurión, Romina Báez, Jaime Vester, Instituto de Investigaciones en Ciencia de la Salud, Universidad Nacional de Asunción, Paraguay; Ana Soilán, Hospital Nacional de Itauguá y Hospital Materno Infantil de San Lorenzo, Ministerio de Salud Pública y Bienestar Social, Asunción, Paraguay; Marina Ortega, Hospital Nacional de Itauguá, e Instituto Nacional del Cáncer, Ministerio de Salud Pública y Bienestar Social, Asunción, Paraguay; Carmen Cristaldo, Hospital Nacional de Itauguá, Ministerio de Salud Pública y Bienestar Social, Asunción, Paraguay. 

Lead author for this group; Valeria Mariel Padín. Email: oncovir@anlis.gob.ar”

Authors´comments to the Editor:

1) Based on Reviewer 1's suggestions, we have added 2 supplementary tables. So, taking into account their order of appearance in the text, we should to renumber the supplementary tables.

2) Some authors are members of the International Agency for Research on Cancer / World Health Organization; for this reason, they have to include a disclaimer statement. Due to we did not know where put it, we decided to add it in ACKNOWLEDGMENTS section, as follows: 

 LINES 441-444: “Where authors are identified as personnel of the International Agency for Research on Cancer / World Health Organization, the authors alone are responsible for the views expressed in this article and they do not necessarily represent the decisions, policy or views of the International Agency for Research on Cancer / World Health Organization.”

---

## [Editor Report · Decision Letter 1]

15 Jul 2022

Distribution of human papillomavirus genotypes by severity of cervical lesions in HPV screened positive women from the ESTAMPA study in Latin America.

PONE-D-22-10439R1

Dear Dr. Correa,

We’re pleased to inform you that your manuscript has been judged scientifically suitable for publication and will be formally accepted for publication once it meets all outstanding technical requirements.

Kind regards,

Maria Lina Tornesello

Academic Editor

PLOS ONE
---

## [Editor Report · Acceptance letter]

20 Jul 2022

PONE-D-22-10439R1 

Distribution of Human Papillomavirus genotypes by severity of cervical lesions in HPV screened positive women from the ESTAMPA study in Latin America.   

Dear Dr. Correa:

I'm pleased to inform you that your manuscript has been deemed suitable for publication in PLOS ONE. Congratulations! Your manuscript is now with our production department. 

Kind regards, 

on behalf of

Dr. Maria Lina Tornesello 

Academic Editor

PLOS ONE